# GeoComplete: Geometry-Aware Diffusion for Reference-Driven Image Completion

**Beibei Lin    Tingting Chen    Robby T. Tan**
National University of Singapore
{beibei.lin, tingting.c}@u.nus.edu, robby.tan@nus.edu.sg

## Abstract

Reference-driven image completion, which restores missing regions in a target view using additional images, is particularly challenging when the target view differs significantly from the references. Existing generative methods rely solely on diffusion priors and, without geometric cues such as camera pose or depth, often produce misaligned or implausible content. We propose **GeoComplete**, a novel framework that incorporates explicit 3D structural guidance to enforce geometric consistency in the completed regions, setting it apart from prior image-only approaches. GeoComplete introduces two key ideas: conditioning the diffusion process on projected point clouds to infuse geometric information, and applying target-aware masking to guide the model toward relevant reference cues. The framework features a dual-branch diffusion architecture. One branch synthesizes the missing regions from the masked target, while the other extracts geometric features from the projected point cloud. Joint self-attention across branches ensures coherent and accurate completion. To address regions visible in references but absent in the target, we project the target view into each reference to detect occluded areas, which are then masked during training. This target-aware masking directs the model to focus on useful cues, enhancing performance in difficult scenarios. By integrating a geometry-aware dual-branch diffusion architecture with a target-aware masking strategy, GeoComplete offers a unified and robust solution for geometry-conditioned image completion. Experiments show that GeoComplete achieves a **17.1%** PSNR improvement over state-of-the-art methods, significantly boosting geometric accuracy while maintaining high visual quality.

## 1  Introduction

Reference-driven image completion restores missing regions in a target image using additional views of the same scene. However, variations in viewpoint, occlusions, dynamic content, and camera settings make it difficult to identify and transfer useful information, posing significant challenges for accurate completion.

To address these challenges, traditional geometry-based methods [35, 48, 49] rely on a sequential pipeline of pose estimation, depth reconstruction, 3D warping, patch fusion, and image harmonization. However, as highlighted in [40], this approach is fragile, as early-stage errors often cascade and lead to failure in complex scenes with occlusions, dynamic content, or ambiguous geometry. To handle complex scenes, generative methods like RealFill [40] fine-tune diffusion models on masked reference images to directly synthesize missing regions. While effective, RealFill struggles when the target view differs significantly from the references. Without geometric cues like camera poses or depth, it often produces hallucinated structures or misaligned completions.

In this paper, we propose GeoComplete, a geometry-aware image completion framework that synthesizes missing regions with strong geometric consistency. GeoComplete is based on two key ideas: (1)

39th Conference on Neural Information Processing Systems (NeurIPS 2025).

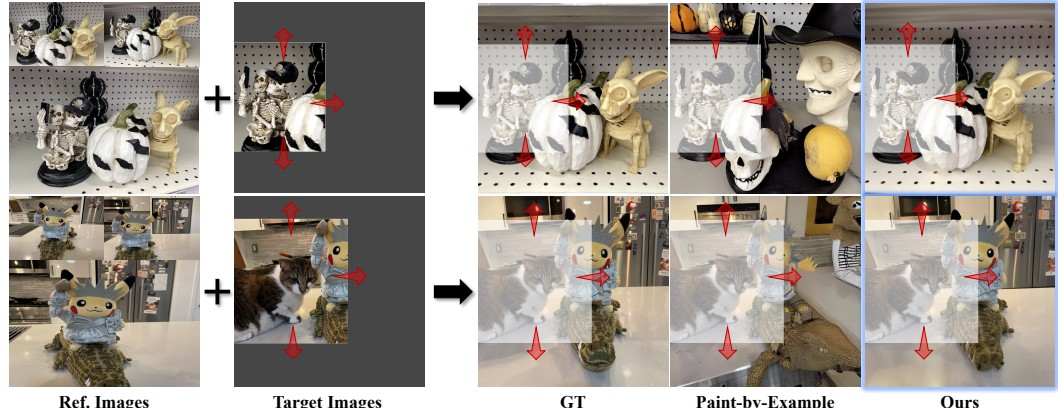

| Ref. Images | Target Images | | GT | Paint-by-Example | Ours |

Figure 1: Given a few reference images of the same scene and a target image with missing regions, our method completes the target's missing regions while preserving geometric consistency more effectively than the state-of-the-art Paint-by-Example [44]. Semi-transparent white masks indicate the known, unaltered regions of the target image.

injecting explicit geometric cues into a diffusion model by conditioning on the projected point cloud, and (2) guiding the model to focus on informative reference regions via target-aware masking. We define "informative regions" as areas that are visible in reference views but missing from the target.

To obtain point clouds, our framework integrates two components: Visual Geometry Grounded Transformer (VGGT) [41] and Language Segment Anything (LangSAM) [25, 22, 32]. Unlike traditional geometry-based methods (e.g., [35, 48, 49]) that rely on sequential estimation steps, VGGT predicts key 3D attributes in a single forward pass. Trained on large-scale data, VGGT delivers accurate and efficient geometry estimation, even in complex scenes, though its performance may degrade with dynamic objects. To address this, we integrate LangSAM, which segments dynamic regions using text prompts. By filtering out moving content before point cloud generation, LangSAM enhances the robustness of geometry estimation. Prompts can be provided manually or generated automatically by a large language model (LLM) [1].

The resulting point cloud is projected to the target view and fed into our dual-branch diffusion framework, comprising a target branch and a cloud branch. The target branch encodes the masked image to generate missing content. The cloud branch processes the projected point cloud to provide geometric guidance. Joint self-attention fuses the two branches, enabling geometry-aware synthesis of missing regions.

To address the challenge of completing regions not visible in the target view, we introduce target-aware masking to guide the model toward useful and non-redundant reference cues. Using 3D attributes from VGGT, we project the target view into each reference to identify informative regions. Rather than masking reference images randomly as in RealFill [40], we selectively mask these informative regions to encourage the model to learn from content that complements the target view.

Figure 1 shows that GeoComplete significantly outperforms state-of-the-art methods, producing missing regions with strong geometric consistency. The main contributions of this work are:

- **Dual-branch Diffusion:** We propose a geometry-aware dual-branch diffusion model that synthesizes missing regions with strong geometric consistency. It comprises a target branch, which conditions the diffusion model on the masked image to generate missing content, and a cloud branch, which conditions it on the projected point cloud to provide geometric cues.

- **Target-aware Masking Strategy:** Unlike RealFill, which applies random masking to the reference image, our method selectively masks informative regions to guide the diffusion model toward meaningful cues, leading to more accurate and coherent completions.

- **Extensive Experiments:** *GeoComplete* significantly outperforms existing methods in both structural accuracy and visual fidelity. Specifically, our method surpasses state-of-the-art approaches by 17.1% in PSNR on benchmark datasets.

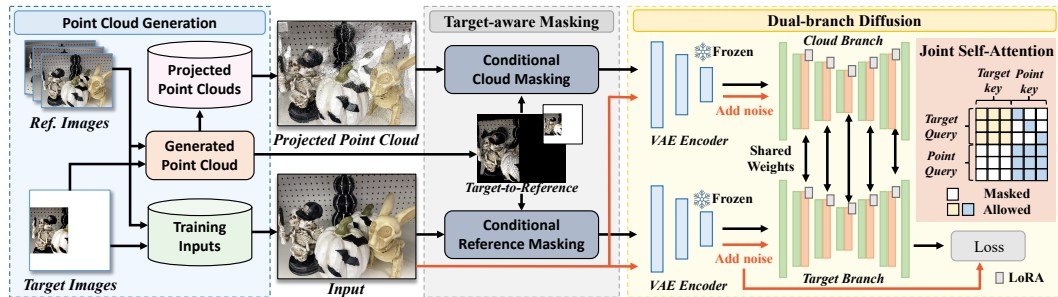

Figure 2: **Overview of our GeoComplete framework:** We first construct a point cloud from the reference and target images. During training, target-aware masking selectively occludes both reference images and their projected point clouds to highlight informative regions. These inputs are processed by a dual-branch diffusion model: the target branch encodes the masked image, while the cloud branch encodes the projected point cloud. Joint self-attention fuses the two, allowing geometric cues to guide synthesis. At inference, the masked target image and its projected point cloud are fed into the finetuned model to complete the missing regions.

## 2 Related Work

**Image Completion** Traditional methods [15, 21, 38, 18] employ task-specific networks to fill missing regions. Existing generative approaches [45, 10, 24, 9, 37, 29, 2, 7, 6, 42, 20] leverage pre-trained diffusion models to achieve strong image generation capabilities. Inspired by this, several methods [37, 29, 2] fine-tune diffusion models with prompt guidance for image completion. In our setting, text prompts fail to capture the rich cues available in reference images, leading to suboptimal results. Reference-driven methods [49, 48, 35] combine depth and pose estimation, image warping, and harmonization, but these components are error-prone and often compound failures, especially in dynamic scenes. Moreover, their limited generative ability hinders plausible content synthesis. Recent diffusion-based methods [44, 40] draw on Stable Diffusion priors. Paint-by-Example [44] uses the target image and a CLIP embedding [28] of a single reference for semantic guidance, while RealFill [40] adapts the diffusion model per scene via LoRA to reconstruct masked references with multiple inputs. However, both approaches neglect geometric cues such as depth and pose, which are crucial for spatial consistency across views. Our method addresses this gap by explicitly injecting geometry into the diffusion model, enabling geometry-aware generation with improved spatial alignment. Concurrently, other works [33, 36] couple NeRF [26] or 3DGS [17, 5] with diffusion for scene inpainting. For example, the Geometric-aware 3D Scene Inpainter [33] conditions diffusion on multi-view images and geometry to reconstruct 3D structure. These methods, however, assume shared geometry across views, limiting applicability in dynamic or varying conditions.

**Geometric Information Estimation** Existing geometry-based completion methods [49, 48, 35] depend on separate estimation modules such as camera pose [46, 16], monocular depth [30, 19, 14], and feature matching with robust fitting [34, 12, 3, 31, 4] to enable view warping. In contrast, the Visual Geometry Grounded Transformer (VGGT) [41] unifies these tasks by jointly predicting camera parameters, depth maps, point maps, and 3D tracks directly from input views. While VGGT achieves strong results in static scenes, it struggles with dynamic objects. To address this, we incorporate LangSAM [25, 22, 32] to filter dynamic content before applying VGGT, enabling more reliable 3D attribute prediction in such settings.

## 3 Proposed Method

Figure 2 shows the overall pipeline of *GeoComplete*, which comprises three key components: point cloud generation, dual-branch diffusion, and target-aware masking. The point cloud generation module estimates camera parameters and depth maps from the reference and target images, constructs a 3D point cloud, and projects it onto both views to provide geometric guidance. The dual-branch diffusion model then synthesizes the missing regions while integrating this geometric information. Finally, the target-aware masking strategy directs the model to focus on reference regions that are not visible from the target view, encouraging the use of complementary cues.

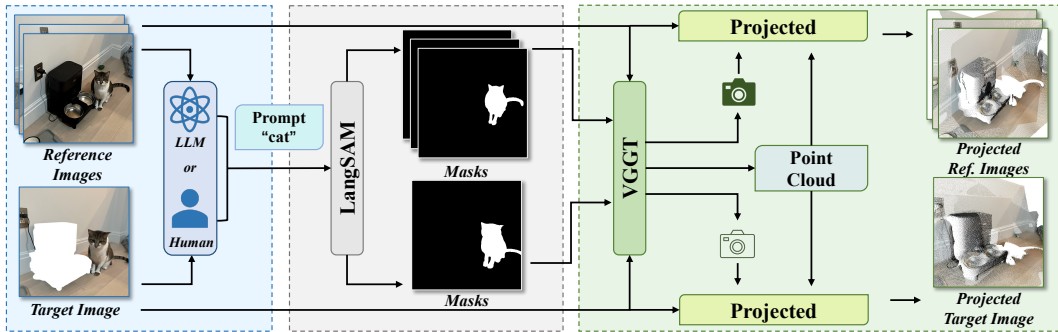

Figure 3: Overview of our point cloud generation pipeline. Given reference and target images, we first obtain a text prompt describing dynamic objects, either provided by users or generated by an LLM [1]. Based on the prompt, LangSAM [25, 22, 32] is employed to segment and remove dynamic regions. VGGT [41] is then applied to estimate camera parameters and depth maps, which are used to construct and project the 3D point cloud.

## 3.1 Point Cloud Generation

Figure 3 shows our point cloud generation pipeline. Given a set of reference images and a target image, we first obtain a text prompt that describes dynamic objects in the scene. The prompt is preferably provided by the user; if unavailable, it is automatically generated by a large language model (LLM)[1]. LangSAM[25, 22, 32] uses this prompt to segment and filter out dynamic regions in both the reference and target images. By removing dynamic objects, geometry estimation focuses on the static scene, enabling reliable correspondences across views.

We use VGGT [41] to jointly estimate camera parameters and depth maps from the filtered reference and target images, avoiding the error accumulation common in multi-stage geometry pipelines. The resulting 3D attributes form point clouds, which are projected onto both views to provide explicit geometric guidance. To prevent over-reliance on potentially inaccurate point clouds, we apply a conditional cloud masking strategy (Section 3.2) that introduces random masking during training.

Given a set of reference images $\{\mathbf{x}_i^{\text{ref}} \mid i = 1, 2, \ldots, N^{\text{ref}}\}$ and a target image $\mathbf{x}^{\text{tar}}$, we introduce a scene-specific text prompt $\mathbf{p}^{\text{dyn}}$ to describe dynamic objects. If not provided by the user, $\mathbf{p}^{\text{dyn}}$ is automatically generated by a large language model (LLM). Using this prompt, we apply LangSAM to segment dynamic regions in both reference and target images. The resulting segmentation masks are $\{\mathbf{m}_i^{\text{ref}}\}$ for the references and $\mathbf{m}^{\text{tar}}$ for the target. We then mask out these dynamic regions to produce filtered images $\{\tilde{\mathbf{x}}_i^{\text{ref}}\}$ and $\tilde{\mathbf{x}}^{\text{tar}}$, preserving only the static content of the scene.

To estimate the camera parameters and depth maps for all filtered reference images and the target image, we formulate the prediction process using VGGT as:

$$\left(\{\mathbf{c}_i^{\text{ref}}\}, \mathbf{c}^{\text{tar}}, \{\mathbf{d}_i^{\text{ref}}\}, \mathbf{d}^{\text{tar}}\right) = f_{\text{vggt}}\left(\{\tilde{\mathbf{x}}_i^{\text{ref}}\}, \tilde{\mathbf{x}}^{\text{tar}}; \theta_{\text{vggt}}\right), \quad (1)$$

where $\{\mathbf{c}_i^{\text{ref}}\}$ and $\{\mathbf{d}_i^{\text{ref}}\}$ are the predicted camera parameters and depth maps for the reference images, and $\mathbf{c}^{\text{tar}}$ and $\mathbf{d}^{\text{tar}}$ are those for the target image. $\theta_{\text{vggt}}$ is the pre-trained parameters of VGGT.

To obtain the projected point cloud for each reference image $\mathbf{x}_i^{\text{ref}}$, we first exclude its own information during point cloud construction. The resulting point cloud is then projected onto the reference view, formulated as:

$$\mathbf{p}_i^{\text{ref}} = \pi\left(\pi^{-1}\left(\{\mathbf{d}_j^{\text{ref}}, \mathbf{c}_j^{\text{ref}} \mid j \neq i\} \cup \{\mathbf{d}^{\text{tar}}, \mathbf{c}^{\text{tar}}\}\right), \mathbf{c}_i^{\text{ref}}\right), \quad (2)$$

where $\pi^{-1}(\cdot)$ denotes the back-projection from depth maps to 3D space, and $\pi(\cdot)$ denotes the forward projection onto the 2D image plane. $\mathbf{p}_i^{\text{ref}}$ is the projected point cloud for the reference image $\mathbf{x}_i^{\text{ref}}$. Similarly, the point cloud constructed from all reference images is projected onto the target view:

$$\mathbf{p}^{\text{tar}} = \pi\left(\pi^{-1}\left(\{\mathbf{d}_j^{\text{ref}}, \mathbf{c}_j^{\text{ref}} \mid \forall j\}\right), \mathbf{c}^{\text{tar}}\right), \quad (3)$$

where $\mathbf{p}^{\text{tar}}$ is the projected point cloud for the target image $\mathbf{x}^{\text{tar}}$.

## 3.2 Target-aware Masking

During training, we apply target-aware masking to selectively mask both the reference images and their projected point clouds. Using 3D geometric attributes from point cloud generation, we project the target image into each reference view to identify regions that are absent in the target (i.e., informative regions). As shown in Figure 2, these informative regions provide complementary information, while the remaining areas are treated as redundant cues.

We then apply two conditional masking strategies: conditional reference masking and conditional cloud masking. The reference masking randomly masks informative regions while preserving redundant ones, encouraging the model to learn from complementary content. The cloud masking, on the other hand, randomly applies white padding to the projected point maps while keeping informative regions intact, guiding the model to leverage geometric cues in these informative areas.

Given the predicted depth maps ($\{\mathbf{d}_i^{\mathrm{ref}}\}$ and $\mathbf{d}^{\mathrm{tar}}$) and camera parameters ($\{\mathbf{c}_i^{\mathrm{ref}}\}$ and $\mathbf{c}^{\mathrm{tar}}$), we project the filtered target image $\tilde{\mathbf{x}}^{\mathrm{tar}}$ onto each reference view $\tilde{\mathbf{x}}_i^{\mathrm{ref}}$, defined as:

$$\mathbf{p}_i^{\mathrm{tar}\to\mathrm{ref}} = \pi\left(\pi^{-1}\left(\mathbf{d}^{\mathrm{tar}}, \mathbf{c}^{\mathrm{tar}}\right), \mathbf{c}_i^{\mathrm{ref}}\right), \tag{4}$$

where $\mathbf{p}_i^{\mathrm{tar}\to\mathrm{ref}}$ is the projection of the target view into the $i$-th reference view. We convert this projection into a binary mask $\mathbf{r}_i^{\mathrm{ref}}$, where visible regions are set to 0 and zero-valued regions are set to 1.

The conditional reference masking is defined as:

$$\hat{\mathbf{x}}_i^{\mathrm{ref}} = \mathbf{x}_i^{\mathrm{ref}} \odot \left((1 - \mathbf{r}_i^{\mathrm{ref}}) + \mathbf{r}_i^{\mathrm{ref}} \odot \mathbf{m}_i^{\mathrm{rand}}\right), \tag{5}$$

where $\mathbf{m}_i^{\mathrm{rand}} \in \{0, 1\}^{H \times W}$ is a random binary mask applied only to the informative regions, and $H$ and $W$ are the height and width of the reference image. The operator $\odot$ denotes element-wise multiplication, and $\hat{\mathbf{x}}_i^{\mathrm{ref}}$ is the resulting masked reference image. This operation preserves redundant content while randomly masking informative, non-redundant regions.

In contrast, conditional cloud masking retains non-redundant geometric regions while applying random masking to redundant ones. It is defined as:

$$\begin{aligned}
\mathbf{m}_i^{\mathrm{point}} &= \mathbf{r}_i^{\mathrm{ref}} + (1 - \mathbf{r}_i^{\mathrm{ref}}) \odot \mathbf{m}_i^{\mathrm{rand}}, \\
\hat{\mathbf{p}}_i^{\mathrm{ref}} &= \mathbf{p}_i^{\mathrm{ref}} \odot \mathbf{m}_i^{\mathrm{point}} + v_{\mathrm{fill}} \times (1 - \mathbf{m}_i^{\mathrm{point}}),
\end{aligned} \tag{6}$$

where $v_{\mathrm{fill}}$ is a predefined fill value assigned to masked-out points, and $\hat{\mathbf{p}}_i^{\mathrm{ref}}$ is the masked projected point cloud for the $i$-th reference view. This operation guides the model to rely on geometric cues in regions where visual reference content is lacking.

Following the training strategy of [40], we also sample the target image $\mathbf{x}^{\mathrm{tar}}$ as input during training. We apply random masking to obtain $\hat{\mathbf{x}}^{\mathrm{tar}}$, while the projected point cloud $\mathbf{p}^{\mathrm{tar}}$ is directly used as $\hat{\mathbf{p}}^{\mathrm{tar}}$.

## 3.3 Dual-branch Diffusion

Figure 2 shows the pipeline of our dual-branch diffusion model, which consists of a target branch and a cloud branch. The target branch conditions the diffusion model on a masked image to generate the missing regions. The cloud branch conditions it on a projected point cloud to provide geometric cues. The masked image and projected point cloud are first encoded into the latent space using a VAE encoder. The resulting latent features, along with a noisy latent, are then passed to a UNet for denoising.

To enable information exchange between branches, we concatenate hidden features from the target and cloud branches before computing self-attention. This design allows the model to adaptively integrate visual and geometric cues. However, since most regions in the target image are masked, the resulting latent features often lack meaningful information. Although these masked tokens can attend to the cloud branch, they struggle to extract useful guidance. To overcome this, we modify the attention mask to explicitly link each masked token in the target branch to its corresponding token in the cloud branch. This ensures that the target branch receives direct geometric cues, even when visual information is absent.

Table 1: Quantitative comparisons on the RealBench benchmark. We evaluate both prompt-based and reference-based inpainting methods across low-level (PSNR, SSIM, LPIPS), mid-level (DreamSim), and high-level (DINO, CLIP) metrics. Higher PSNR, SSIM, DINO, CLIP, and User Study scores (ranging from 1 to 5), and lower LPIPS and DreamSim scores, indicate better performance. We highlight the best and second-best results for each metric.

| Method | | RealBench | | | | | | QualBench |
| | | Low-level | | | Mid-level | High-level | | User |
| | | PSNR ↑ | SSIM ↑ | LPIPS ↓ | DreamSim ↓ | DINO ↑ | CLIP ↑ | Study ↑ |
|---|---|---|---|---|---|---|---|---|
| Prompt-based | SD Inpaint | 10.63 | 0.282 | 0.605 | 0.213 | 0.831 | 0.874 | 2.13 |
| | Generative Fill | 10.92 | 0.311 | 0.598 | 0.212 | 0.851 | 0.898 | 2.61 |
| Reference-based | Paint-by-Example | 10.13 | 0.244 | 0.642 | 0.237 | 0.797 | 0.859 | 1.85 |
| | TransFill | 13.28 | 0.404 | 0.542 | 0.192 | 0.860 | 0.866 | – |
| | RealFill | 14.78 | 0.424 | 0.431 | 0.077 | 0.948 | 0.962 | 3.98 |
| | Ours | 17.32 | 0.578 | 0.197 | 0.036 | 0.986 | 0.987 | 4.61 |

Given the hidden features from the target and cloud branches, denoted as $\mathbf{h}_{\text{tar}}$ and $\mathbf{h}_{\text{pt}}$, we concatenate them along the token dimension:

$$\mathbf{h}_{\text{cat}} = \text{Concat}(\mathbf{h}_{\text{tar}}, \mathbf{h}_{\text{pt}}), \tag{7}$$

where $\mathbf{h}_{\text{cat}} \in \mathbb{R}^{2L \times d}$, with $L$ representing the number of tokens per branch and $d$ the feature dimension. The combined features $\mathbf{h}_{\text{cat}}$ are then used for self-attention.

To control information flow, we introduce an attention mask $\mathbf{m}_{\text{attn}} \in \mathbb{R}^{2L \times 2L}$ during self-attention. The mask is constructed to: (1) allow tokens within the same branch to attend to each other, (2) permit each target-branch token to attend to its corresponding cloud-branch token, and (3) block all other cross-branch interactions. An illustration of the attention mask is shown in Figure 2. The joint self-attention is formulated as:

$$\mathbf{h}_{\text{attn}} = f_{\text{self-attn}}(\mathbf{h}_{\text{cat}}, \mathbf{m}_{\text{attn}}), \tag{8}$$

where $f_{\text{self-attn}}(\cdot)$ denotes the masked self-attention operation, and $\mathbf{h}_{\text{attn}}$ is the resulting attended feature.

During training, the diffusion loss is defined as:

$$\mathcal{L} = \frac{1}{B} \sum_{j=1}^{B} \mathbb{E}_{t,\epsilon} \left[ \|\mathbf{w}_j \cdot (\epsilon - \epsilon_\theta(\mathbf{x}_j(t), t, \hat{\mathbf{p}}_j, \hat{\mathbf{x}}_j))\|_2^2 \right], \tag{9}$$

where $\mathcal{L}$ is the diffusion loss, $B$ is the batch size, and $\epsilon_\theta(\cdot)$ denotes the predicted noise. The conditional inputs satisfy $\hat{\mathbf{x}}_j \in \{\hat{\mathbf{x}}_i^{\text{ref}}\} \cup \{\hat{\mathbf{x}}^{\text{tar}}\}$ and $\hat{\mathbf{p}}_j \in \{\hat{\mathbf{p}}_i^{\text{ref}}\} \cup \{\hat{\mathbf{p}}^{\text{tar}}\}$. Here, $\mathbf{x}_j(t)$ is the ground-truth image at timestep $t$, and $\mathbf{w}_j$ is a weighting map indicating valid regions (e.g., visible areas in the target view). The loss is computed only over these valid regions.

During inference, we use $\mathbf{p}^{\text{tar}}$ and $\mathbf{x}^{\text{tar}}$ as conditional inputs to guide the dual-branch diffusion, generating missing regions while preserving geometric structures.

# 4 Experiments

In our experiments, we follow the evaluation protocol of [40] and test on two challenging reference-based image completion datasets: RealBench and QualBench.

**RealBench** [40] contains 33 scenes (23 outpainting and 10 inpainting). Each scene provides 1–5 reference images, a target image with missing regions, a binary mask, and a ground-truth completion. Scenes include large variations between target and references, such as viewpoint, blur, lighting, style, and pose. Evaluation uses six metrics: PSNR, SSIM, LPIPS [47], DreamSim [13], DINO [8], and CLIP [28]. PSNR, SSIM, and LPIPS capture low-level quality, while DreamSim, DINO, and CLIP assess perceptual fidelity at mid- and high-levels.

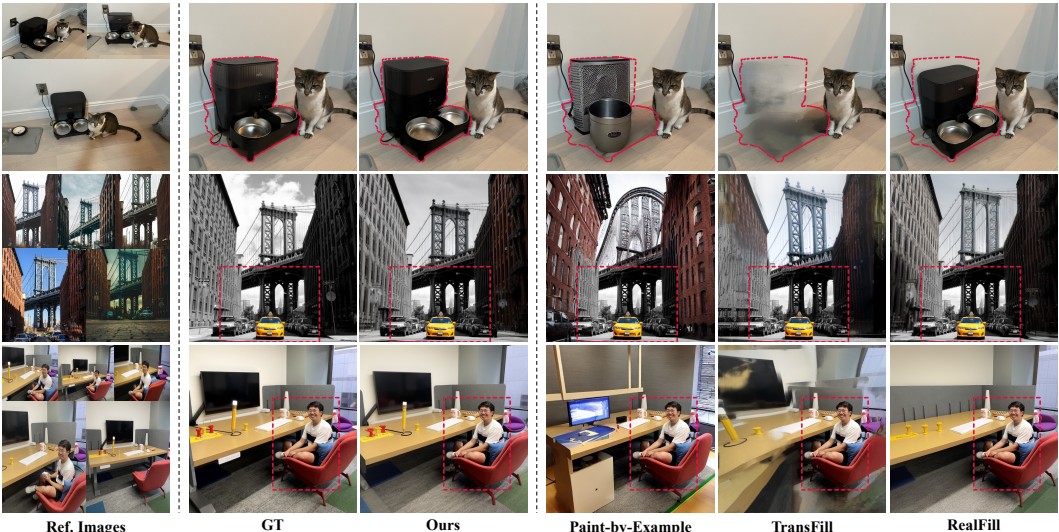

| Ref. Images | GT | Ours | Paint-by-Example | TransFill | RealFill |

Figure 4: Qualitative comparisons from Transfill [49], RealFill [40], Paint-by-Example [44] and our method. The red bounding box marks the known, unaltered region of the target image (i.e., the area inside the box), except for the first-row images, where the known region lies outside the box. Our method synthesizes missing regions while ensuring better geometric consistency.

**QualBench** [40] includes 25 scenes collected in the same way but without ground-truth completions. We therefore conduct a user study where participants rate each result (1–5) based on: (1) realism of the restored content, (2) consistency with references, and (3) structural and color coherence with the unmasked target. Higher scores reflect more natural, geometrically consistent, and visually coherent completions.

## 4.1 Implementation Details

All experiments are conducted on a server equipped with four NVIDIA GPUs, each with 24 GB of memory. Our implementation involves three key components: point cloud generation, target-aware masking, and dual-branch diffusion, each of which is described in detail below.

**Point Cloud Generation** Our point cloud generation pipeline incorporates two key components: LangSAM [25, 22, 32] and VGGT [41]. In LangSAM, we employ SAM 2.1-Large [32] for segmentation. The text prompts are either manually provided by users or automatically generated by a large language model, ChatGPT-4o [27]. Since VGGT only supports inputs of size $518 \times 518$, we resize the reference and target images while preserving their aspect ratios. After resizing, a center crop is applied to obtain the final $518 \times 518$ resolution.

**Target-aware Masking** Our target-aware masking consists of a conditional reference masking and a conditional point masking. Following the strategy in [38, 39], the conditional reference masking first generates multiple random rectangles and constructs the initial mask by either taking their union or the complement of their union. Subsequently, following Equation 5, it selectively unmasks less informative regions in the reference images. Similarly, the conditional point masking first generates the initial mask and then selectively unmasks non-redundant geometric regions, as defined in Equation 6. The fill value $v_{\text{fill}}$ is set to 1 (white) to replace masked-out regions in the projected point cloud.

**Dual-branch Diffusion** Our diffusion model is built upon Stable Diffusion 2 Inpainting [37]. We fine-tune it with LoRA, updating only rank-decomposed layers in the U-Net while keeping original weights frozen. The LoRA rank is set to 8 to balance adaptation capacity and training efficiency. For each scene, we fine-tune the model for 2,000 iterations with a batch size of 16. During training, all reference and target images are resized to a resolution of $512 \times 512$.

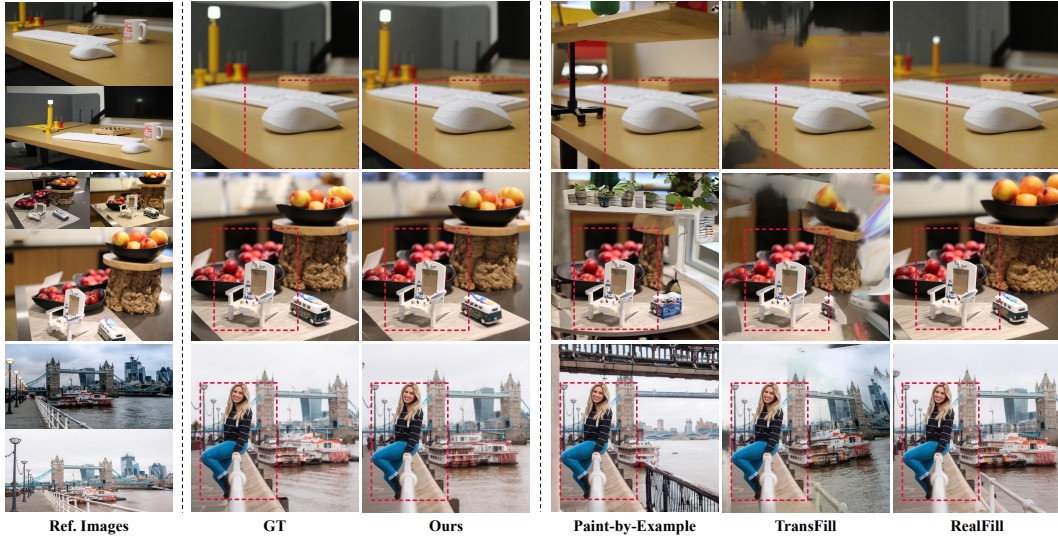

| Ref. Images | GT | Ours | Paint-by-Example | TransFill | RealFill |

Figure 5: Qualitative comparisons from Transfill [49], RealFill [40], Paint-by-Example [44] and our method. The red bounding box marks the known, unaltered region of the target image (i.e., the area inside the box). Our method synthesizes missing regions while ensuring better geometric consistency.

## 4.2 Evaluation

**Quantitative** Table 1 compares GeoComplete with state-of-the-art methods on RealBench and QualBench. Baselines include prompt-based approaches (SD Inpaint [37], Generative Fill [2]) and reference-based methods (Paint-by-Example [44], TransFill [49], RealFill [40]). The prompt-based models rely on text input, while the reference-based ones use images for completion.

Compared to prompt-based methods, *GeoComplete* achieves large gains across all low-level metrics, improving PSNR by over 5 dB and reducing LPIPS from 0.605 to 0.237. Against TransFill, a geometry-aware baseline, our model benefits from VGGT and LangSAM to generate more reliable point clouds, yielding notable improvements. While RealFill already performs strongly with masked reference conditioning, *GeoComplete* further improves SSIM (0.424 → 0.555) and reduces LPIPS by 0.194, producing sharper and more perceptually faithful reconstructions. These results underscore the importance of explicit 3D geometric priors and validate the effectiveness of our dual-branch diffusion design.

**Qualitative** Figures 4 and 5 show qualitative comparisons on RealBench. Generative frameworks such as RealFill leverage reference images to produce plausible completions, but without explicit geometry they often fail to maintain spatial consistency, leading to misaligned or implausible content. In contrast, *GeoComplete* enforces geometric consistency by integrating priors from LangSAM and VGGT within a dual-branch architecture that jointly encodes visual and 3D cues. As illustrated in Figure 4, *GeoComplete* reconstructs fine details and preserves scene-level alignment, even under large viewpoint changes between the target and references.

## 4.3 Ablation Studies

We conduct ablation experiments to evaluate the contributions of target-aware masking and dual-branch diffusion. Results are reported in Table 2. Without geometric guidance, our method reduces to RealFill, shown in the first row as the baseline.

**Dual-branch Diffusion** We compare GeoComplete with and without explicit geometric cues. As shown in Table 2, removing geometry causes clear drops across all metrics (e.g., PSNR and SSIM decrease by 1.59 and 0.131). Figures 4 and 5 further illustrate that without geometry, RealFill often produces hallucinated or misaligned content. By contrast, GeoComplete integrates geometric information into the generation process, yielding structurally consistent results. This highlights the importance of explicit geometry in guiding diffusion-based restoration.

Table 2: Ablation study on the effectiveness of dual-branch diffusion, joint self-attention, and target-aware masking. We report low-level (PSNR, SSIM, LPIPS), mid-level (DreamSim), and high-level (DINO, CLIP) metrics. Higher PSNR, SSIM, DINO, and CLIP scores and lower LPIPS and DreamSim scores indicate better performance.

| Dual-branch Diffusion | Joint Self-Attention with Mask | Target-aware Masking | Low-level | | | Mid-level | High-level | |
|---|---|---|---|---|---|---|---|---|
| | | | PSNR ↑ | SSIM ↑ | LPIPS ↓ | DreamSim ↓ | DINO ↑ | CLIP ↑ |
| × | × | × | 14.78 | 0.424 | 0.431 | 0.077 | 0.948 | 0.962 |
| ✓ | × | × | 16.37 | 0.555 | 0.237 | 0.049 | 0.981 | 0.982 |
| ✓ | ✓ | × | 16.85 | 0.564 | 0.219 | 0.045 | 0.983 | 0.984 |
| ✓ | ✓ | ✓ | **17.32** | **0.578** | **0.197** | **0.036** | **0.986** | **0.987** |

Table 3: Robustness of GeoComplete to VGGT and LangSAM Errors. We simulate (1) noisy point clouds, (2) sparse point clouds, and (3) LangSAM segmentation errors. "0% / 25% / 50% / 75%" indicate ratios of points perturbed or removed. For the LangSAM case, "w/." and "w/o." denote using or removing the masks, while "+Rand." denotes randomly adding 10% extra masked regions per mask. CM = Conditional Cloud Masking, JSA = Joint Self-Attention. PSNR (dB) is reported.

| Method | Noisy Point Cloud | | | | Sparse Point Cloud | | | | LangSAM (13 scenes) | | |
|---|---|---|---|---|---|---|---|---|---|---|---|
| | 0% | 25% | 50% | 75% | 0% | 25% | 50% | 75% | w/. | w/o. | +Rand. |
| RealFill | 14.78 | 14.78 | 14.78 | 14.78 | 14.78 | 14.78 | 14.78 | 14.78 | 14.44 | 14.44 | 14.44 |
| Ours w/o. CM & JSA | 16.37 | 14.60 | 14.51 | 14.35 | 16.37 | 14.58 | 14.50 | 14.35 | 15.92 | 14.54 | 14.58 |
| Ours | **17.32** | **17.14** | **17.03** | **16.90** | **17.32** | **17.18** | **16.83** | **16.50** | **16.83** | **16.66** | **16.51** |

**Joint Self-Attention** We ablate the joint self-attention module, which fuses target and cloud-branch features under a controlled attention mask. As shown in Table 2, removing this module results in noticeable drops in PSNR, CLIP, and DINO, reflecting weaker low-level fidelity and high-level semantic alignment. These results demonstrate the role of joint self-attention in ensuring alignment between the target and projected point cloud.

**Target-aware Masking** We also evaluate the effect of target-aware masking. Removing this strategy consistently reduces performance across all metrics (e.g., PSNR and SSIM drop by 0.47 and 0.014, while CLIP and DINO decrease by 0.003). This indicates that target-aware masking helps the model focus on non-redundant regions in the references, improving inference accuracy and fidelity.

**Robustness of GeoComplete** GeoComplete relies on upstream modules such as VGGT and LangSAM, whose outputs may contain errors (see Sec. 3.1). To mitigate this, we introduce conditional cloud masking (CM), which prevents the model from over-relying on unreliable geometry. We also employ joint self-attention with masking (JSA), which enforces token-to-token links between the target and cloud branches, ensuring that noisy cloud tokens do not propagate globally through cross-attention, particularly when they dominate.

To evaluate robustness, we simulate (1) noisy point clouds, (2) sparse point clouds, and (3) LangSAM segmentation errors. Details can be found in the Appendix. As shown in Table 3, GeoComplete shows only small drops under these perturbations and consistently outperforms both RealFill and the variant without CM and JSA. These results demonstrate the effectiveness of CM and JSA in maintaining strong performance even when upstream predictions are noisy or partially erroneous.

## 5 Conclusion

We introduced *GeoComplete*, a geometry-guided diffusion framework for reference-driven image completion. Unlike existing generative methods that operate solely in the image domain, GeoComplete incorporates explicit 3D geometry by conditioning the diffusion model on projected point clouds. To guide the model toward meaningful reference cues, we propose a target-aware masking strategy that filters redundant content and emphasizes complementary regions. Our dual-branch architecture jointly processes geometric and visual tokens through self-attention, enabling the synthesis of structurally accurate and visually coherent results. Extensive experiments on real-world and synthetic benchmarks demonstrate that GeoComplete achieves clear improvements over state-of-the-art methods in both geometric consistency and perceptual quality.

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

# Appendix

## A Implementation Details

### A.1 Detailed Clarification of Workflow

Initially, we process all reference images and the target image using VGGT and LangSAM to obtain the point cloud.

During training, given a reference image $I_{ref}$ and its corresponding projected point cloud $I_{cloud}$, we use our conditional reference masking to mask the reference image, obtaining the augmented reference image $I_{aug-ref} \in \mathbb{R}^{3 \times H \times W}$ and the mask image $I_{mask} \in \mathbb{R}^{1 \times H \times W}$. We also use our conditional cloud masking to augment the projected point cloud, resulting in $I_{aug-cloud} \in \mathbb{R}^{3 \times H \times W}$. We then encode these images into the latent space using a VAE. This results in a latent reference image $I_{ref}^{latent}$ (used as the ground truth), a latent masked reference $I_{aug-ref}^{latent}$, and a latent projected point cloud $I_{aug-cloud}^{latent}$, all in $\mathbb{R}^{4 \times h \times w}$, where $h = H/8$ and $w = W/8$. The mask is also downsampled to $I_{mask}^{latent} \in \mathbb{R}^{1 \times h \times w}$.

To fine-tune the diffusion model, we add noise to the ground truth latent $I_{ref}^{latent}$ to obtain a noisy latent $I_{noisy}^{latent}$. The input to the target branch is the concatenation of $I_{noisy}^{latent}$, $I_{mask}^{latent}$, and $I_{aug-ref}^{latent}$, resulting in a tensor of shape $\mathbb{R}^{9 \times h \times w}$. Similarly, the input to the cloud branch is the concatenation of $I_{noisy}^{latent}$, $I_{mask}^{latent}$, and $I_{aug-cloud}^{latent}$, also in $\mathbb{R}^{9 \times h \times w}$. The objective is to estimate the added noise, which has shape $\mathbb{R}^{4 \times h \times w}$.

During inference, given the target image $I_{tar}$, its corresponding projected point cloud $I_{cloud}$, and the mask image $I_{mask}$, we directly process them into the latent space. This results in a latent target $I_{tar}^{latent}$ and a latent projected point cloud $I_{cloud}^{latent}$. The mask is also downsampled to $I_{mask}^{latent}$. We initialize $I_{noisy}^{latent}$ using standard Gaussian noise. Then, we concatenate the corresponding latent tensors to construct the inputs for the target and cloud branches. After an iterative denoising process and using the VAE decoder, we obtain the final output.

### A.2 Details of Baseline Methods

For SD Inpaint [37] and Generative Fill [2], we follow the instructions from RealFill to generate long descriptions for each scene with the help of ChatGPT. For RealFill [40], we follow their official setting by fixing the text prompt to a sentence containing a rare token, i.e., "a photo of [V]". For a fair comparison, our method also adopts this setting.

### A.3 Robustness Evaluation Details

To evaluate robustness, we simulate three conditions that introduce errors from VGGT and LangSAM:

1. **Noisy Point Cloud:** Gaussian noise is added to a subset of points in the generated 3D point cloud to mimic degraded geometry.
2. **Sparse Point Cloud:** A ratio of points is randomly dropped from the 3D point cloud before projection to simulate sparse geometry.
3. **Segmentation Errors:** We manually selected 13 RealBench scenes with significant dynamic objects and tested two variants: (1) removing LangSAM masks entirely and (2) introducing errors by randomly adding 10% extra masked regions per mask.

### A.4 Prompt Design

Since dynamic objects can significantly affect the geometric predictions of VGGT [41], we introduce LangSAM [25, 22, 32] to filter out dynamic content before applying VGGT, thereby enabling robust 3D attribute prediction even in dynamic scenes. Prompts can be provided manually or generated automatically by a large language model (LLM) [1].

When using an LLM, we upload all reference images along with the target image. The following guided prompt is used:

Table 4: Comparison with existing reference-guided image generation methods. Results are reported on different scene subsets (13 for Step1X-Edit, 28 for OmniGen, and 28 for Bagel). PSNR and SSIM are reported.

| Method | 13 scenes (Step1x-Edit) | | 28 scenes (OmniGen) | | 28 scenes (Bagel) | |
|---|---|---|---|---|---|---|
| | PSNR | SSIM | PSNR | SSIM | PSNR | SSIM |
| Step1X-Edit | 9.95 | 0.3678 | – | – | – | – |
| OmniGen | – | – | 8.93 | 0.3525 | – | – |
| Bagel | – | – | – | – | 10.83 | 0.4705 |
| RealFill | 15.75 | 0.5130 | 14.92 | 0.5156 | 14.92 | 0.5043 |
| Ours | 18.12 | 0.5869 | 17.37 | 0.5857 | 17.48 | 0.5827 |

```
Identify and list only the objects that are inconsistent across the
images, such as dynamic objects that change position, appearance, or
are missing.  Ignore consistent background objects even if the viewpoint
changes slightly.
```

## B    Comparison with Reference-Guided Image Editing Methods

In this section, we evaluate existing reference-guided image editing methods, including OmniGen [43], Step1x-Edit [23], and Bagel [11]. We initially follow the official instructions to run these baseline models. For scenes where the models fail to perform adequately, we employ ChatGPT to generate prompts and manually refine them as needed. However, these methods fail to handle all scenes. In some cases, the restored results become completely white or visually meaningless. In summary, only 28 scenes from OmniGen, 13 scenes from Step1X-Edit, and 28 scenes from Bagel produce valid outputs. PSNR and SSIM are computed on these successfully restored results, as summarized in Table 4. Overall, the results suggest that under the reference-based image completion setting, existing reference-guided image generation methods still perform suboptimally.

## C    Computational Cost

Table 5 summarizes the computational cost (using four 24G GPUs) of Paint-by-Example [44], RealFill [40], and our method. Both RealFill and our method are based on per-scene optimization. For a fair comparison, we adopt identical experimental settings, including batch size, number of optimization steps, and number of GPUs. Although our approach introduces slightly higher overhead than RealFill, it achieves significantly better reconstruction quality. Notably, our method reaches promising results within 500 steps (18 mins), outperforming RealFill even at 2000 steps (50 mins).

Compared to one-shot models such as Paint-by-Example, per-scene optimization methods generally provide more accurate content restoration but are less suitable for time-sensitive or real-time applications. As shown in Figure 4 and Figure 5, our method produces more faithful results, while Paint-by-Example often fails to preserve fine-grained content from the reference images.

To explore potential acceleration strategies, we first identify that the primary computational bottleneck in our framework lies in the 2,000-step per-scene fine-tuning process. One potential solution is to pre-train the LoRA parameters of the diffusion model on a large-scale, task-specific dataset. This would serve as a strong initialization for subsequent per-scene adaptation, thereby significantly reducing the number of required optimization steps while preserving the quality of generated results. We consider this an important direction for future work.

## D    Limitations

While GeoComplete effectively leverages geometric cues for reference-driven image completion, it inherits certain limitations from its components. For example, the quality of the projected point cloud depends on the accuracy of the geometry estimation module (e.g., VGGT [41]). To mitigate the impact of inaccurate point clouds, we introduce a conditional cloud masking strategy that prevents

Table 5: Computational cost and performance comparison. We report pre-processing overhead, training time, inference time, and PSNR (dB).

| Method | Pre-processing (one time) | Training Time | Inference | PSNR |
|---|---|---|---|---|
| Paint-by-Example | None | One-shot | ∼30s | 10.13 |
| RealFill [40] (500 steps) | None | 12 mins | ∼8s | 13.67 |
| RealFill [40] (2000 steps) | None | 48 mins | ∼8s | 14.78 |
| Ours (500 steps) | VGGT + LangSAM <30s | 18 mins | ∼15s | 16.33 |
| Ours (2000 steps) | VGGT + LangSAM <30s | 72 mins | ∼15s | 17.32 |

the model from relying on unreliable geometric input. This allows our framework to generate realistic results even when the point cloud is inaccurate. However, when the point cloud is imprecise, the framework may not be able to fully exploit geometric information, which can affect completion quality in those regions.

# E  Societal Impact

Reference-driven image completion can benefit various applications, including occlusion removal, image editing, and scene understanding in both consumer and industrial domains. GeoComplete introduces explicit geometric information to guide the completion process, reducing hallucination risks and improving structural fidelity. However, as the framework relies on generative models to synthesize missing content, it may still produce plausible yet inaccurate completions, especially in regions with limited geometric or visual cues. Therefore, we recommend caution when applying such methods in safety-critical or forensic contexts that require guaranteed factual accuracy.

