# OpenReview forum: "GeoComplete: Geometry-Aware Diffusion for Reference-Driven Image Completion"
_NeurIPS.cc/2025/Conference — NeurIPS 2025 poster_

### Official Review · Reviewer_qNjF · 2025-06-25

**Clarity:** 3
**Significance:** 3
**Originality:** 3
**Rating:** 4
**Confidence:** 3

**Summary:**

This paper introduces a reference-guided image outpainting method by introducting feed-forward 3D foundation models to faciliate 2D tasks. It predicts geometric-aware outpainted regions through conditioning on the a 3D point cloud, and finetunes or use off-the-shelf diffusion models to fill in the missing region. It is an interesting and reasonable idea in itself that has demonstrated good results over the baseline. The evaluation on quality is sufficient.

**Questions:**

How would a language model at all be able to tell what is the dynamic region of the image? I don't think the paper clearly defines it either. Considering the example on the first page, the cat was not in the reference images, how is it considered by the language model then?

How is the runtime of this model compare to baseline and especially RealFill?

**Ethical Concerns:**

["NO or VERY MINOR ethics concerns only"]

**Final Justification:**

I keep my initial rating for the paper as borderline accept. The rebuttal provided by the authors addressed my concerns sufficiently.

**Limitations:**

yes

**Paper Formatting Concerns:**

This paper has no formatting issue.

**Quality:**

3

**Strengths And Weaknesses:**

In general the idea makes sense, as filling in a 2d capture of a 3d scene with the reference from the 3D content will naturally outperform pure 2d based models in terms of the persective of the surroundings. My only concern is the runtime overload this method, which I believe should be heavier than 2D based methods which only requires model finetuning. It would be nice if the author can acknowledge this in the limitation or provide justification otherwise.

---

> ### Author Rebuttal · Authors · 2025-07-31
>
> ### **[W1/Q2: Runtime Comparisons]**
> The Table A below summarizes the computational cost (using four 24G GPUs) of Paint-by-Example [32], RealFill [30], and our method. Both RealFill and our method are based on per-scene optimization. For a fair comparison, we adopt identical experimental settings, including batch size, number of optimization steps, and number of GPUs. Although our approach introduces slightly higher overhead than RealFill, it achieves significantly better reconstruction quality. Notably, our method reaches promising results within 500 steps (18 minutes), outperforming RealFill even at 2000 steps (50 minutes).
>
> Compared to one-shot models such as Paint-by-Example, per-scene optimization methods generally provide more accurate content restoration but are less suitable for time-sensitive or real-time applications. As shown in Figures 4 and 5, our method produces more faithful results by leveraging scene-specific optimization, whereas Paint-by-Example often fails to transfer fine-grained content from the reference images due to the absence of such tuning.
>
> *Table A. Computational Cost and Performance Comparison.*
> | Method    | Pre-processing (one time)        | Training Time | Inference | PSNR    |
> |-----------|------------------------|----------------|-----------|---------|
> | Paint-by-Example  | None           | One-shot       | ～30s        | 10.13   |
> | RealFill (500 steps)  | None                   | 12 mins        | ～8s        | 13.67 	    |
> | RealFill (2000 steps) | None                   | 48 mins        | ～8s        | 14.78   |
> | Ours  (500 steps)    | VGGT + LangSAM < 30s   | 18 mins        | ～15s        | 16.33 |
> | Ours  (2000 steps)    | VGGT + LangSAM < 30s   | 72 mins        | ～15s        | **17.32** |
>
> ---
>
> ### **[Q1: Clarification on Dynamic Regions]**
> To identify dynamic regions, we use an LLM (GPT-4o) with the target image and all reference images as input, prompting it to detect any dynamic content. The exact prompt used is provided in our supplementary material (Lines 8–11).
>
> We acknowledge that LangSAM may fail, particularly in highly complex scenes. Such failures can degrade the quality of the generated masks and, consequently, the reconstructed point clouds. However, our method does not heavily rely on the projected point cloud for image completion. Instead, we introduce two key components: Conditional Cloud Masking (CM) and the Joint Self-Attention (JSA) module with masking, which are designed to prevent the model from over-relying on potentially noisy or inaccurate geometric information.
>
> During training, although the projected point cloud may already contain inaccuracies (e.g., due to VGGT), the conditional cloud masking module further masks portions of it. Supervised by ground-truth RGB images, the model learns that not all point cloud information is reliable, allowing it to adaptively interpret and utilize geometric cues under challenging conditions.
>
> In addition, our joint self-attention module with masking (JSA) can potentially mitigate this issue. It establishes explicit token-to-token links between the target and cloud branches. This design prevents noisy cloud tokens from globally influencing all target tokens through cross-attention, especially when noisy tokens are more numerous.
>
> To support our claim, we present the following experiments for verification. We manually select 13 scenes with significant dynamic objects from RealBench to evaluate the robustness of our method. We conduct two experiments: (1) removing the Langsam masks entirely, and (2) simulating segmentation errors by randomly adding 10% additional masked regions to each Langsam mask. The results are shown in Table B. When Langsam fails, our method exhibits only a slight drop of approximately 0.3 dB in PSNR. In contrast, the variant without CM and JSA shows a substantially larger drop of approximately 1.4 dB, highlighting the robustness of our full method to segmentation errors introduced by Langsam.
>
> *Table B. PSNR under Different Langsam Mask Conditions on Dynamic Scenes.*
> | Methods                   | w/. langsam  | w/o. langsam| langsam + random mask|
> |---------------------------|-------------|------------|------------|
> | RealFill                  | 14.44       | 14.44      |14.44|
> | Ours w/o. CM & JSA        | 15.92       | 14.54 (↓1.38)     |14.58 (↓1.34)|
> | Ours                      | 16.83       | 16.66  (↓0.17)    |16.51 (↓0.32)|

---

> > ### Comment · Reviewer_qNjF · 2025-08-05
> >
> > Thanks to the authors' additional justification. I keep my initial rating as borderline accept. Good luck!

---

> > > ### Author Response · Authors · 2025-08-05
> > >
> > > Thank you for reviewing our paper. We sincerely appreciate the time and effort you’ve dedicated to evaluating our work and for your thoughtful and positive feedback. We just wanted to kindly check if you could update your final rating, as it appears it has not yet been entered.

---

### Official Review · Reviewer_VDvq · 2025-07-03

**Clarity:** 3
**Significance:** 3
**Originality:** 3
**Rating:** 5
**Confidence:** 4

**Summary:**

The authors propose a new method for image completion (inpainting, outpainting, etc.) given (a) an image to be completed and (b) several other reference views of the scene, from different viewpoints. This longstanding problem in computer vision has been worked on even since before the days of DNNs, and work continues into the diffusion age. This paper suggests using a state-of-the-art point cloud method to do (sparse) novel viewpoint generation from the reference images, then trains an image completer that can consume these projected pointclouds. They make experimental comparisons to RealFill and Paint-By-Example.

**Questions:**

How do comparisons with Transfill work?

How does the model degrade, as compared to other methods' degradation, as there is less overlap, i.e., the pointcloud covers less and less of the image and the model is forced to do more and more whole-cloth extrapolation?

**Ethical Concerns:**

["NO or VERY MINOR ethics concerns only"]

**Final Justification:**

I remain in support of accepting this paper; I'm happy that the authors will address some less-than-clear parts of the paper.

**Limitations:**

yes

**Paper Formatting Concerns:**

lgtm

**Quality:**

3

**Strengths And Weaknesses:**

Quality: Qualitatively and quantitatively the results are convincing. It seems that TransFill is a more apt comparison than PBE, especially given the paper’s claims about the superiority of one-shot geometry (lines 27-30), so I would prefer for the paper to include some qualitative comparisons here, especially ones that show clearly why pointclouds are a better 3D representation than multiple homographies.

Clarity: the paper is fairly readable and appropriately concise. This reviewer was a little bit confused about how the conditioning was added to the diffusion model; it looks at first like the conditioning itself has noise added; it’s hard to see how this could work at inference time since the noise, at least in traditional diffusion levels, would destroy all the conditioning signal! At first I thought it might be some kind of CFM variant  – but now I think the latents of the conditioning are concatenated onto the noise (lines 167-168).

Originality and significance: this is a novel approach that seems to improve the SOTA on a long-standing and well-established vision problem.

I note that the authors promise to release code and training scripts, which is great from a reproducibility standpoint.

(edited to add:) the authors may also be interested in comparing with this work: https://arxiv.org/pdf/2502.13335 (I believe the work to be roughly simultaneous).

---

> ### Author Rebuttal · Authors · 2025-07-31
>
> ### **[W1: One-shot Geometry vs. Multiple Homographies]**
> The limitations of multiple homographies have been previously identified and evidenced by RealFill [30], which demonstrates catastrophic failure when the scene’s structure cannot be accurately estimated. In contrast, extensive experiments in VGGT validate the superiority of one-shot geometry estimation. Due to this year’s rebuttal policy, we are unable to include additional comparison figures. However, some TransFill results are already available in the RealFill paper [30]. We also commit to including direct qualitative comparisons with TransFill in the revised version.
>
> ---
>
> ### **[W2: Clarification of Workflow]**
> Initially, we process all reference images and the target image using VGGT and Langsam to obtain the point cloud.
>
> During training, given a reference image $I_{ref}$ and its corresponding projected point cloud $I_{cloud}$, we use our conditional reference masking to mask the reference image, obtaining the augmented reference image $I_{aug-ref} \in \mathbb{R}^{3 \times H \times W}$ and the mask image $I_{mask} \in \mathbb{R}^{1 \times H \times W}$. We also use our conditional cloud masking to augment the projected point cloud, resulting in $I_{aug-cloud} \in \mathbb{R}^{3 \times H \times W}$. We then encode these images into the latent space using a VAE. This results in a latent reference image $I_{ref}^{latent}$ (used as the ground truth), a latent masked reference $I_{aug-ref}^{latent}$, and a latent projected point cloud $I_{aug-cloud}^{latent}$, all in $\mathbb{R}^{4 \times h \times w}$, where $h = H / 8$ and $w = W / 8$. The mask is also downsampled to $I_{mask}^{latent} \in \mathbb{R}^{1 \times h \times w}$.
>
>
> To fine-tune the diffusion model, we add noise to the ground truth latent $I_{ref}^{latent}$ to obtain a noisy latent $I_{noisy}^{latent}$. The input to the target branch is the concatenation of $I_{noisy}^{latent}$, $I_{mask}^{latent}$, and $I_{aug-ref}^{latent}$, resulting in a tensor of shape $\mathbb{R}^{9 \times h \times w}$. Similarly, the input to the cloud branch is the concatenation of $I_{noisy}^{latent}$, $I_{mask}^{latent}$, and $I_{aug-cloud}^{latent}$, also in $\mathbb{R}^{9 \times h \times w}$. The objective is to estimate the added noise, which has shape $\mathbb{R}^{4 \times h \times w}$.
>
> During inference, given the target image $I_{tar}$, its corresponding projected point cloud $I_{cloud}$, and the mask image $I_{mask}$, we directly process them into the latent space. This results in a latent target $I_{tar}^{latent}$ and a latent projected point cloud $I_{cloud}^{latent}$. The mask is also downsampled to $I_{mask}^{latent}$. We initialize $I_{noisy}^{latent}$ using standard Gaussian noise. Then, we concatenate the corresponding latent tensors to construct the inputs for the target and cloud branches. After an iterative denoising process and using the VAE decoder, we obtain the final output.
>
>
>
>
>
> ---
>
> ### **[W3: Comparison with Recent Work]**
> Following the suggestion, we provide a discussion comparing our method with Geometric-aware 3D Scene Inpainter [R1]. This method conditions a diffusion model on multi-view images and scene geometry to reconstruct accurate 3D structure, which in turn supports high-quality inpainting. However, it assumes that all multi-view inputs share the same underlying scene geometry, which limits its applicability under challenging conditions such as dynamic objects or varying lighting. In contrast, our approach uses 3D geometry only as auxiliary guidance, rather than a reconstruction target. By leveraging the generative capabilities of the diffusion model, our method focuses on completing missing image regions with greater flexibility and robustness, even when the geometric cues are imperfect.
>
> [R1] Salimi, Ahmad, et al. "Geometry-aware diffusion models for multiview scene inpainting." arXiv preprint arXiv:2502.13335 (2025).
>
> ---
>
> ### **[Q1: Clarification of TransFill [37]]**
> The quantitative results of TransFill are directly taken from the RealFill [30] paper, and we follow the same evaluation setting to ensure a fair comparison. Since the official code of TransFill is not publicly available, we, like the RealFill team, contacted the authors to obtain the qualitative results.
>
> ---
>
> ### **[Q2: Robustness of GeoDiff]**
>
> Following the suggestions, we conduct ablation studies on the scenario with reduced overlap. As the point cloud covers less of the image, the number of points decreases, leading to reduced pixel information in the projected point cloud. To simulate this scenario, we perform the following experiments:
>
> 1) **Pixel Drop in the Projected Point Cloud:** We randomly drop pixels within the target regions of the projected point cloud, directly reducing the available point cloud information. The experimental results are presented in Table A.
>
> 2) **Region Drop in the Projected Point Cloud:** We randomly remove multiple rectangular blocks within the target regions of the projected point cloud until the total number of removed points reaches a fixed ratio of the points in these regions. This block-based removal simulates region-level information loss. The experimental results are presented in Table B.
>
> 3) **Sparse Point Cloud Simulation:** We randomly drop a certain ratio of points from the point cloud before projection to simulate sparse projected point clouds. The experimental results are in Table C.
>
> It can be observed that our method remains more robust under these conditions, consistently outperforming the baseline method RealFill. This robustness arises from the fact that our method does not heavily rely on the projected point cloud for image completion. Instead, we introduce two key components: Conditional Cloud Masking (CM) and the Joint Self-Attention (JSA) module with masking, which are designed to prevent the model from over-relying on potentially noisy or inaccurate geometric information.
>
> During training, although the projected point cloud may already contain inaccuracies (e.g., due to VGGT), the conditional cloud masking module further masks portions of it. Supervised by ground-truth RGB images, the model learns that not all point cloud information is reliable, allowing it to adaptively interpret and utilize geometric cues under challenging conditions.
>
> In addition, our joint self-attention module with masking (JSA) can potentially mitigate this issue. It establishes explicit token-to-token links between the target and cloud branches. This design prevents noisy cloud tokens from globally influencing all target tokens through cross-attention, especially when noisy tokens are more numerous.
>
> Furthermore, in Tables A, B, and C, we present the ablation studies for our CM and JSA modules. When the point cloud or projected pixels become sparse, even with only 25% of the points dropped, the variant without CM and JSA exhibits a significant performance drop. In contrast, our full method remains robust under these conditions.
>
> *Table A. PSNR under Different Drop Ratios of Pixels in the Projected Point Cloud.*
>
> | Method               | Original   |  Drop 25%  Pixels    | Drop 50%   Pixels    | Drop 75%    Pixels   |
> |----------------------|--------------|-------------|-------------|-------------|
> | RealFill             | 14.78       | 14.78       | 14.78       | 14.78       |
> | Ours w/o. CM & JSA | 16.37       | 14.55 (↓1.82)      | 14.45 (↓1.92)      | 14.29 (↓2.08)       |
> | Ours           | **17.32**    | 16.85 (↓0.47)      | 16.51 (↓0.81)       | 15.76 (↓1.56)      |
>
> *Table B. PSNR under Different Drop Ratios of Regions in the Projected Point Cloud.*
> | Method               | Original   |  Drop 25%  Regions    | Drop 50%   Regions    | Drop 75%    Regions   |
> |----------------------|--------------|-------------|-------------|-------------|
> | RealFill             | 14.78       | 14.78       | 14.78       | 14.78       |
> | Ours w/o. CM & JSA | 16.37       | 14.67 (↓1.7)      | 14.65 (↓1.72)      | 14.56  (↓1.81)       |
> | Ours           | **17.32**    | 17.08 (↓0.24)      | 16.83 (↓0.49)       | 16.61  (↓0.71)      |
>
> *Table C. PSNR under Different Drop Ratios of Points in the Generated 3D Point Cloud.*
> | Method               | Original   | Drop 25%  Points    | Drop 50%   Points    | Drop 75%    Points   |
> |----------------------|-------------|----------------|----------------|----------------|
> | RealFill             | 14.78       | 14.78          | 14.78          | 14.78          |
> | Ours w/o. CM & JSA | 16.37       | 14.58 (↓1.79)  | 14.50 (↓1.87)  | 14.35 (↓2.02)  |
> | Ours                 | **17.32**   | 17.18 (↓0.14)  | 16.83 (↓0.35)  | 16.50 (↓0.82)  |

---

> > ### Comment · Reviewer_VDvq · 2025-08-07
> >
> > Thanks, that method clarification is helpful. I suggest incorporating it into the paper. The robustness numbers are also impressive. I remain in support of acceptance.
> >
> > One thing to note in reading through the other reviewers' work, you write:
> >
> > """
> > To our knowledge, GeoDiff is the first to tightly couple explicit 3D geometry with diffusion-based image completion in a unified framework.
> > """
> >
> > which 4jUZ has called out. I was also startled by this claim (and should have noted it in my initial review, my oversight, sorry). Obviously given my accept rating I think there is novelty here, but I would strongly urge you to temper that claim in a final revision; I agree with 4jUZ that it is probably overbroad as written.

---

> > > ### Author Response · Authors · 2025-08-07
> > >
> > > Thank you for the suggestion and for supporting our paper. We will incorporate the method clarification in the final version. Regarding the statement highlighted by Reviewer 4jUZ, we agree that the phrasing should be tempered, particularly around the use of geometry as a prior. Our key contributions lie in the geometry-aware dual-branch diffusion architecture and the target-aware masking strategy, which together offer a unified and robust solution for geometry-conditioned image completion.

---

> > > ### Author Response · Authors · 2025-08-07
> > >
> > > Dear Reviewer VDvq,
> > >
> > > We would like to kindly follow up to check if you might be able to update the mandatory acknowledgment for the rebuttal, as it appears it has not been entered yet. Additionally, if you have already made a decision on the final rating, we noticed it has not been submitted either.
> > >
> > > Thank you again for your support of our paper. We truly appreciate it, and we are happy to answer any further questions you may have.
> > >
> > > Sincerely,
> > >
> > > Authors

---

### Official Review · Reviewer_vSDB · 2025-07-03

**Clarity:** 3
**Significance:** 2
**Originality:** 3
**Rating:** 4
**Confidence:** 3

**Summary:**

This paper presents GeoDiff, a framework for reference-driven image completion that explicitly leverages 3D geometry to guide a diffusion-based generation process. The work targets a key limitation of existing generative methods: when there is a significant viewpoint discrepancy between the reference and target views, current approaches often yield geometrically inconsistent or misaligned results due to the absence of explicit 3D priors. The proposed method demonstrates impressive visual quality and delivers substantial improvements over prior state-of-the-art baselines, both quantitatively and qualitatively.

**Questions:**

While the paper presents promising results, I still have a few unresolved concerns. I would be willing to re-evaluate my score upon receiving satisfactory clarification from the authors.
1. How robust is the method to errors from upstream modules such as LangSAM and VGGT?
The proposed pipeline depends on accurate segmentation and depth/pose estimation from external models. Have the authors evaluated how errors in these stages (e.g., incorrect masks or noisy point clouds) affect the final image completion results? Are there any mechanisms to mitigate error propagation?
2. Can the authors elaborate on the computational cost and scalability of the approach?
Given that the method requires 2,000 iterations of per-scene fine-tuning in addition to inference from LangSAM and VGGT, how practical is the framework for large-scale or real-time applications? Have the authors explored any strategies for reducing the computational burden?
3. Models such as OmniGen, Step1x-Edit, and Bagel also support reference-guided image generation and could serve as strong baselines. Could the authors clarify the rationale behind the choice of baselines, and possibly include comparisons with these more recent methods?

**Ethical Concerns:**

["NO or VERY MINOR ethics concerns only"]

**Final Justification:**

Most of my concerns have been addressed, and I will be increasing my rating accordingly. Additionally, I strongly encourage the authors to update their final draft based on the rebuttal (Computational Cost & Potential Baselines).

**Limitations:**

Please refer Questions for details.

**Quality:**

2

**Strengths And Weaknesses:**

**Strengths**
1. Novelty and Significance. A key strength of the paper lies in its novel integration of explicit 3D geometric priors into a diffusion-based framework for reference-guided image completion. This design directly addresses a major limitation of prior generative approaches—namely, the inability to maintain geometric consistency under large viewpoint changes—and achieves demonstrable improvements in geometric fidelity.

2. Good results. The paper conducts comprehensive empirical validation, including both standard quantitative metrics (PSNR, SSIM, LPIPS, etc.) and a user study for qualitative assessment. The ablation studies are well-designed and effectively highlight the individual contributions of each model component.

**Weaknesses**
1. Dependency on Upstream Modules. The framework's reliance on external modules such as LangSAM (for segmentation) and VGGT (for depth and pose estimation) introduces a significant vulnerability. Inaccuracies in these modules can propagate through the pipeline and compromise the quality of geometric guidance, ultimately degrading the final output. However, the paper does not sufficiently analyze or quantify these potential failure modes.

2. High Computational Cost.The proposed approach requires per-scene fine-tuning of the diffusion model over 2,000 iterations, alongside the overhead of running LangSAM and VGGT. This makes the method computationally demanding and limits its applicability in time-sensitive or real-time settings. A discussion on scalability or potential acceleration strategies would be beneficial.

3. Limited Handling of Dynamic and Complex Scenes. The reliance on text-based prompts in LangSAM introduces limitations in dynamic or cluttered environments, especially when accurate or complete textual descriptions are unavailable. This restricts the method’s generalizability to complex, real-world scenes with multiple or moving objects.

4. Lack of Comparison with Stronger Baselines. The experimental section omits comparison with several recent multi-modal generation methods, such as OmniGen[1], Step1x-Edit[2], and Bagel[3]. Including such baselines would provide a more complete picture of the method's relative advantages and limitations, especially since these models also support reference-conditioned generation in diverse settings.

References:
- [1] OmniGen: Unified Image Generation
- [2] Step1X-Edit: A Practical Framework for General Image Editing
- [3] Unified Model for Multimodal Understanding and Generation

---

> ### Author Rebuttal · Authors · 2025-07-31
>
> ### **[W1/Q1: Robustness for Upstream Modules]**
> We acknowledge that during both training and inference, the upstream modules (LangSAM and VGGT) may consequently generate inaccurate projected point clouds, as noted in Lines 115–116.
>
> To address this issue, we introduce a conditional cloud masking (CM) strategy (Lines 150–159) that prevents the model from over-relying on potentially noisy or erroneous geometry. During training, although the projected point cloud may already contain inaccuracies from VGGT, we further mask portions of it. With supervision from ground-truth RGB images, the model learns that not all information from point clouds is trustworthy, enabling it to adaptively interpret noisy geometry cues.
>
> In addition, our joint self-attention module with masking (JSA) can potentially mitigate this issue. It establishes explicit token-to-token links between the target and cloud branches. This design prevents noisy cloud tokens from globally influencing all target tokens through cross-attention, especially when noisy tokens are more numerous.
>
> Consequently, at inference time, even when VGGT or LandSAM outputs are inaccurate, our model can still adaptively integrate information from both projected point clouds and RGB signals to produce reliable results.
>
> To support our claim, we present the following experiments for verification.
> 1) **Noisy Point Cloud Simulation:** To simulate degraded or poor-quality geometry, we randomly add Gaussian noise to a subset of points in the generated 3D point cloud. The experimental results are shown in Table A. Although the performance of our method slightly degrades with increasing noise levels, it consistently outperforms the baseline method RealFill across all settings.
>
> *Table A. PSNR under Different Ratios of Noisy Points in the Generated 3D Point Cloud.*
> | Method               |  Original   | 25% Noisy Points   | 50% Noisy Points    |  75% Noisy  Points  |
> |----------------------|-------------|----------------|----------------|----------------|
> | RealFill             | 14.78       | 14.78          | 14.78          | 14.78          |
> | Ours w/o. CM & JSA | 16.37       | 14.60  (↓1.77)  | 14.51 (↓1.86)  | 14.35 (↓2.02)  |
> | Ours                 | **17.32**   | 17.14  (↓0.18)  |  17.03 (↓0.29)  | 16.90  (↓0.42)  |
>
> 2) **Sparse Point Cloud Simulation:** We randomly drop a certain ratio of points from the point cloud before projection to simulate sparse projected point clouds. The experimental results are in Table B. It can be observed that when the point cloud becomes sparse, even with only 25% of points dropped, the method without CM and JSA experiences a significant performance drop. In contrast, our full method remains more robust under these conditions.
>
> *Table B. PSNR under Different Drop Ratios of Points in the Generated 3D Point Cloud.*
> | Method               | Original   | Drop 25%  Points    | Drop 50%   Points    | Drop 75%    Points   |
> |----------------------|-------------|----------------|----------------|----------------|
> | RealFill             | 14.78       | 14.78          | 14.78          | 14.78          |
> | Ours w/o. CM & JSA | 16.37       | 14.58 (↓1.79)  | 14.50 (↓1.87)  | 14.35 (↓2.02)  |
> | Ours                 | **17.32**   | 17.18 (↓0.14)  | 16.83 (↓0.35)  | 16.50 (↓0.82)  |
>
> 3) **Ablation Study on LangSAM:** We manually select 13 scenes with significant dynamic objects from RealBench to evaluate the robustness of our method. We conduct two experiments: (1) removing the Langsam masks entirely, and (2) simulating segmentation errors by randomly adding 10% additional masked regions to each Langsam mask. The results are shown in Table C below. When Langsam fails, our method exhibits only a slight drop of approximately 0.3 dB in PSNR. In contrast, the variant without CM and JSA shows a substantially larger drop of approximately 1.4 dB, highlighting the robustness of our full method to segmentation errors introduced by Langsam.
>
> *Table C. PSNR under Various Langsam Mask Conditions for Scenes with Challenging Dynamic Objects.*
> | Methods                   | w/. langsam  | w/o. langsam| langsam + random mask|
> |---------------------------|-------------|------------|------------|
> | RealFill                  | 14.44       | 14.44      |14.44|
> | Ours w/o. CM & JSA        | 15.92       | 14.54 (↓1.38)     |14.58 (↓1.34)|
> | Ours                      | 16.83       | 16.66  (↓0.17)    |16.51 (↓0.32)|
>
> ---
>
> ### **[W2/Q2: Computational Cost and Potential Acceleration Strategies]**
> The Table D below summarizes the computational cost (using four 24G GPUs) of Paint-by-Example [32], RealFill [30], and our method. Both RealFill and our method are based on per-scene optimization. For a fair comparison, we adopt identical experimental settings, including batch size, number of optimization steps, and number of GPUs. Although our approach introduces slightly higher overhead than RealFill, it achieves significantly better reconstruction quality. Notably, our method reaches promising results within 500 steps (18 mins), outperforming RealFill even at 2000 steps (50 mins).
>
> Compared to one-shot models such as Paint-by-Example, per-scene optimization methods generally provide more accurate content restoration but are less suitable for time-sensitive or real-time applications. As shown in Figures 4 and 5, our method produces more faithful results, while Paint-by-Example often fails to preserve fine-grained content from the reference images.
>
> To explore potential acceleration strategies, we first identify that the primary computational bottleneck in our framework lies in the 2,000-step per-scene fine-tuning process. One potential solution is to pre-train the LoRA parameters of the diffusion model on a large-scale, task-specific dataset. This would serve as a strong initialization for subsequent per-scene adaptation, thereby significantly reducing the number of required optimization steps while preserving the quality of generated results. We consider this an important direction for future work.
>
> *Table D. Computational Cost and Performance Comparison.*
> | Method    | Pre-processing (one time)        | Training Time | Inference | PSNR    |
> |-----------|------------------------|----------------|-----------|---------|
> | Paint-by-Example  | None           | One-shot       | ～30s        | 10.13   |
> | RealFill (500 steps)  | None                   | 12 mins        | ～8s        | 13.67 	    |
> | RealFill (2000 steps) | None                   | 48 mins        | ～8s        | 14.78   |
> | Ours  (500 steps)    | VGGT + LangSAM < 30s   | 18 mins        | ～15s        | 16.33 |
> | Ours  (2000 steps)    | VGGT + LangSAM < 30s   | 72 mins        | ～15s        | **17.32** |
>
> ---
>
> ###  **[W3: Dynamic and Complex Scenes]**
> We acknowledge that LangSAM may struggle in dynamic or cluttered environments, especially when accurate or complete textual prompts are unavailable. These limitations can affect the quality of the generated masks and, consequently, the reconstructed point clouds. However, our method treats the 3D point cloud as an auxiliary input rather than a rigid intermediate representation. While the geometric cues from the point cloud may be imperfect, our model selectively leverages reliable information through a conditional cloud masking mechanism and a joint self-attention module. This enables the model to focus on trustworthy regions while ignoring potentially misleading geometry.
>
> Moreover, we would like to emphasize that the benchmark datasets RealBench and QualBench encompass a wide range of complex scenes, including significant dynamic content, multiple moving objects, and varying lighting conditions. The corresponding experimental results are presented in the Table C. These results demonstrate the strong robustness and generalization ability of our method, even in dynamic and complex scenarios.
>
> ---
>
> ### **[W4/Q3.1: Potential Baselines]**
> Following the suggestion, we evaluate existing reference-guided image generation methods, including OmniGen, Step1x-Edit, and Bagel. We initially follow the official instructions to run these baseline models. For scenes where the models fail to perform adequately, we employ ChatGPT to generate prompts and manually refine them as needed. However, these methods fail to handle all scenes. In some cases, the restored results become completely white or visually meaningless. In summary, only 28 scenes from OmniGen, 13 scenes from Step1X-Edit, and 28 scenes from Bagel produce valid outputs. PSNR and SSIM are computed only on these successfully restored results. The results suggest that under the reference-based image completion setting, existing reference-guided image generation methods still perform suboptimally.
>
> | 13 scenes  | PSNR  | SSIM   | |  28 scenes  | PSNR  | SSIM   |
> |-----------|-------|--------| --| ----------|-------|--------|
> | Step1X-Edit    | 9.95  | 0.3678 |  | OmniGen   | 8.93  | 0.3525 |
> | RealFill  | 15.75 | 0.5130 |  | RealFill  | 14.92 | 0.5156 |
> | Ours      | 18.12 | 0.5869 |  | Ours      | 17.37 | 0.5857 |
>
> | 28 scenes | PSNR  | SSIM   |
> |-----------|-------|--------|
> | Bagel     | 10.83 | 0.4705 |
> | RealFill  | 14.92 | 0.5043 |
> | Ours      | 17.48 | 0.5827 |
>
> ---
>
> ### **[Q3.2: Rationale of Baselines]**
> Our baseline methods are primarily selected from the RealFill paper [30], which serves as a strong and recent benchmark in this field. Specifically, we include prompt-based approaches such as SD Inpaint [27] and Generative Fill [2], as well as reference-based completion methods like Paint-by-Example [32], TransFill [37], and RealFill [30]. Notably, both TransFill and RealFill are trained on dedicated image completion datasets, making them particularly suitable for comparison. Overall, these baselines constitute a representative and diverse set of state-of-the-art techniques, enabling fair and meaningful evaluation.

---

> > ### Comment · Reviewer_vSDB · 2025-08-05
> >
> > Thank you to the authors for the rebuttal and discussion. Most of my concerns have been addressed, and I will be increasing my rating accordingly.
> > Additionally, I strongly encourage the authors to update their final draft based on the rebuttal (Computational Cost & Potential Baselines).

---

> > > ### Author Response · Authors · 2025-08-05
> > >
> > > Thank you for your positive feedback. We are glad to hear that your major concerns have been addressed. We are committed to updating the Computational Cost and Potential Baselines experiments and analyses in our final manuscript based on the rebuttal. We would also like to kindly check if you could update the final rating, as it appears that it has not yet been entered.

---

### Official Review · Reviewer_4jUZ · 2025-07-07

**Clarity:** 2
**Significance:** 2
**Originality:** 1
**Rating:** 2
**Confidence:** 3

**Summary:**

This paper introduces GeoDiff, a framework for reference-driven image completion, a task that aims to fill missing regions in a target image using information from other views of the same scene. The authors identify a critical weakness in existing state-of-the-art generative methods, such as RealFill, which, despite their powerful image priors, often produce geometrically inconsistent or misaligned results due to a lack of explicit 3D awareness. GeoDiff proposes to remedy this by directly integrating 3D geometric information into the generative process. Empirically, the authors demonstrate substantial improvements over existing methods on the RealBench and QualBench datasets (which are also introduced in this paper).

**Questions:**

* Given the existence of several peer-reviewed works on geometry-aware generative in-painting from 2022-2025, could you please clarify and refine your paper's core novelty claim beyond being the "first" to couple geometry and diffusion for this task?

* The framework relies heavily on the output of VGGT. Could you discuss the robustness of GeoDiff to potential failures in the geometry estimation stage? Specifically, how does the model perform when provided with a noisy or inaccurate point cloud, and have you considered adding an ablation study to demonstrate this?

* Could you please expand on the evaluation setting for the baseline methods (e.g., how were the prompts chosen for SD, RealFill, etc)?

**Ethical Concerns:**

["NO or VERY MINOR ethics concerns only"]

**Limitations:**

yes

**Quality:**

1

**Strengths And Weaknesses:**

Strengths
 * The paper tackles an important problem. Methods like RealFill have pushed the state of the art by personalizing diffusion models on a few reference images, but their reliance on a 2D image prior without geometric understanding often leads to structural failures, hallucinations, and misalignments when there are significant viewpoint changes between the reference and target images. GeoDiff's central goal of enforcing explicit 3D geometric consistency is therefore not just an incremental improvement but a necessary and logical next step for building more robust, reliable, and authentic image completion tools. Solving this problem has direct applications in computational photography, virtual reality, and digital content creation.

* Well-motivated technical approach that leverages VGGT and Lang-SAM pre-trained models.

* Compelling empirical evaluation -- the paper presents a comprehensive set of experiments that, on the surface, convincingly demonstrate the method's effectiveness.

Weaknesses:
 * Incomplete literature review. The failure to cite and discuss these works is not a minor oversight. It creates a cascade of problems: it allows for an inflated and inaccurate novelty claim, it prevents a meaningful comparison against the true state-of-the-art in geometry-aware generative inpainting, and it ultimately weakens the perceived contribution of the paper. Some important omissions include:
   *  GeoFill (WACV 2023), explicitly estimates a 3D mesh and relative camera pose from two views to guide reference-based inpainting. While it does not use a diffusion model for the final synthesis, it establishes a strong and direct precedent for leveraging explicit 3D scene representation for this task.
   * Geometry-Aware Diffusion Models for Multiview Scene Inpainting. It introduces a "geometry-aware conditional diffusion model" that uses projected geometric and appearance cues from reference images to perform multi-view consistent inpainting.
   * OccludeNeRF focuses on 3D NeRF inpainting by guiding a diffusion model with collaborative score distillation. It explicitly reasons about geometry and occlusions to achieve 3D consistent results, representing another instance of combining 3D scene representations with diffusion models for completion tasks.

* QualBench & RealBench - are not standard benchmarks. The paper introduces these two datasets, instead of re-using a dataset from a prior work, e.g., RealFill has one available. Generally, it's not a good idea to evaluate a new method on a new (not a well established) dataset.

---

> ### Author Rebuttal · Authors · 2025-07-31
>
> ### **[W1: Literature Review]**
> Thank you for highlighting these relevant works. Below, we clarify the differences between our method and prior approaches:
> - GeoFill: We would like to clarify that our manuscript already discusses GeoFill [36] in lines 24–30 and 75–79. In particular, as noted in the RealFill [30] paper, GeoFill heavily relies on accurate 3D scene representations, making it sensitive to reconstruction errors. Moreover, RealFill highlights several limitations of GeoFill, including its difficulty in handling complex scene geometry, appearance changes, and scene deformation. In contrast, our method treats the 3D point cloud as an auxiliary input rather than a rigid intermediate representation. While the geometry cues from the point cloud may be imperfect, our model selectively leverages reliable information through a conditional cloud masking mechanism and a joint self-attention module. This design enhances robustness and enables more accurate and flexible image completion across diverse scenarios.
>
> - Geometric-aware 3D Scene Inpainter: This method conditions a diffusion model on multi-view images and scene geometry to reconstruct accurate 3D structure, which in turn supports high-quality inpainting. However, it assumes that all multi-view inputs share the same underlying scene geometry, which limits its applicability under challenging conditions such as dynamic objects or varying lighting. In contrast, our approach uses 3D geometry only as auxiliary guidance, rather than a reconstruction target. By leveraging the generative capabilities of the diffusion model, our method focuses on completing missing image regions with greater flexibility and robustness, even when the geometric cues are imperfect.
>
> - OccludeNeRF: OccludeNeRF distills knowledge from a pre-trained diffusion model for 3D scene inpainting using Score Distillation Sampling. In contrast, our method directly conditions the diffusion model on projected point clouds and target cues to complete missing regions. While both approaches incorporate geometry within a diffusion framework, the underlying mechanisms and task objectives are fundamentally different.
>
> We note that Geometry-aware 3D Scene Inpainter (2025.02)  and OccludeNeRF (2025.04) are very recent arXiv preprints released around the same time as our submission. We will include the above discussion to clarify the distinctions between these works and ours in the revised version.
>
> ---
>
> ### **[W2: QualBench & RealBench]**
> We would like to clarify that both QualBench and RealBench are not newly created datasets, but directly derived from the RealFill dataset [30], covering all evaluation data used in RealFill. We follow the same evaluation settings, including all datasets and metrics, as established in RealFill to ensure fair and consistent comparison.
>
> ---
>
> ### **[Q1: Contribution]**
> Beyond integrating diffusion with geometry, **our key innovations lie in the dual-branch diffusion architecture and the target-aware masking strategy, both specifically designed for reference-based image completion.**
>
> Our dual-branch diffusion comprises a target branch and a cloud branch. The target branch conditions the diffusion model on the masked image to generate missing content, while the cloud branch conditions it on the projected point cloud to provide geometric cues. A joint self-attention module is introduced to fuse information from both branches.
>
> This design enables the model to adaptively integrate visual and geometric signals. However, since most regions in the target image are masked, the resulting latent features often lack meaningful content. While masked tokens can attend to the cloud branch, they may still struggle to extract effective guidance.
>
> To address this, we modify the attention mask to explicitly link each masked token in the target branch to its corresponding token in the cloud branch, ensuring that the target branch receives direct geometric cues even in the absence of visual information.
>
> Our target-aware masking guides the model to focus on useful and non-redundant reference cues. It consists of two components: conditional reference masking and conditional cloud masking. Unlike RealFill [30], which randomly masks reference images, conditional reference masking selectively masks informative regions to encourage the model to learn from content that complements the target view. Conversely, conditional cloud masking randomly applies white padding to the projected point maps, masking redundant regions while preserving the informative ones, thereby guiding the model to effectively utilize geometric cues. Additionally, masking the projected point cloud informs the model that not all information from the point cloud is trustworthy. This prevents the model from over-relying on potentially noisy, sparse, or inaccurate geometry, thereby enhancing its robustness.
>
> All these task-specific designs enable our method to selectively leverage geometric cues, even when they are imperfect. This distinguishes our approach from prior work, which often relies on highly accurate 3D reconstructions or treats geometry as a rigid intermediate representation.
>
> ---
>
> ### **[Q2: Robustness of GeoDiff]**
> We agree that during both training and inference, the output of VGGT may contain inaccuracies, as noted in Lines 115–116.
>
> To address this issue, we introduce a conditional cloud masking (CM) strategy (Lines 150–159) that prevents the model from over-relying on potentially noisy or erroneous geometry. During training, although the projected point cloud may already contain inaccuracies from VGGT, we further mask portions of it. With supervision from ground-truth RGB images, the model learns that not all information from point clouds is trustworthy, enabling it to adaptively interpret noisy geometry cues.
>
> In addition, our joint self-attention module with masking (JSA) can potentially mitigate this issue. It establishes explicit token-to-token links between the target and cloud branches. This design prevents noisy cloud tokens from globally influencing all target tokens through cross-attention, especially when noisy tokens are more numerous.
>
> Consequently, at inference time, even when VGGT outputs are inaccurate, our model can still adaptively integrate information from both projected point clouds and RGB signals to produce reliable results.
>
> To support our claim, we present the following experiments for verification.
> 1) **Noisy Point Cloud Simulation:** To simulate degraded or poor-quality geometry, we randomly add Gaussian noise to a subset of points in the generated 3D point cloud. The experimental results are shown in Table A. Although the performance of our method slightly degrades with increasing noise levels, it consistently outperforms the baseline method RealFill across all settings.
>
> *Table A. PSNR under Different Ratios of Noisy Points in the Generated 3D Point Cloud.*
> | Method               |  Original   | 25% Noisy Points   | 50% Noisy Points    |  75% Noisy  Points  |
> |----------------------|-------------|----------------|----------------|----------------|
> | RealFill             | 14.78       | 14.78          | 14.78          | 14.78          |
> | Ours w/o. CM & JSA | 16.37       | 14.60  (↓1.77)  | 14.51 (↓1.86)  | 14.35 (↓2.02)  |
> | Ours                 | **17.32**   | 17.14  (↓0.18)  |  17.03 (↓0.29)  | 16.90  (↓0.42)  |
>
> 2) **Sparse Point Cloud Simulation:** We randomly drop a certain ratio of points from the point cloud before projection to simulate sparse projected point clouds. The experimental results are in Table B. It can be observed that when the point cloud becomes sparse, even with only 25% of points dropped, the method without CM and JSA experiences a significant performance drop. In contrast, our full method remains more robust under these conditions.
>
> *Table B. PSNR under Different Drop Ratios of Points in the Generated 3D Point Cloud.*
> | Method               | Original   | Drop 25%  Points    | Drop 50%   Points    | Drop 75%    Points   |
> |----------------------|-------------|----------------|----------------|----------------|
> | RealFill             | 14.78       | 14.78          | 14.78          | 14.78          |
> | Ours w/o. CM & JSA | 16.37       | 14.58 (↓1.79)  | 14.50 (↓1.87)  | 14.35 (↓2.02)  |
> | Ours                 | **17.32**   | 17.18 (↓0.14)  | 16.83 (↓0.35)  | 16.50 (↓0.82)  |
>
> 3) **Ablation Study on LangSAM:** We manually select 13 scenes with significant dynamic objects from RealBench to evaluate the robustness of our method. We conduct two experiments: (1) removing the Langsam masks entirely, and (2) simulating segmentation errors by randomly adding 10% additional masked regions to each Langsam mask. The results are shown below. When Langsam fails, our method exhibits only a slight drop of approximately 0.3 dB in PSNR. In contrast, the variant without CM and JSA shows a substantially larger drop of approximately 1.4 dB, highlighting the robustness of our full method to segmentation errors introduced by Langsam.
>
> *Table C. PSNR under Various Langsam Mask Conditions for Scenes with Challenging Dynamic Objects.*
> | Methods                   | w/. langsam  | w/o. langsam| langsam + random mask|
> |---------------------------|-------------|------------|------------|
> | RealFill                  | 14.44       | 14.44      |14.44|
> | Ours w/o. CM & JSA        | 15.92       | 14.54 (↓1.38)     |14.58 (↓1.34)|
> | Ours                      | 16.83       | 16.66  (↓0.17)    |16.51 (↓0.32)|
>
> ---
>
> ### **[Q3: Details of Baseline Methods]**
> For SD Inpaint [27] and Generative Fill [2], we follow the instructions from RealFill, which generate long descriptions for each scene with the help of ChatGPT. For RealFill [30], we follow their official setting by fixing the text prompt to a sentence containing a rare token, i.e., “a photo of [V]”. For a fair comparison, our method also adopts this setting.

---

> ### Author Response · Authors · 2025-08-06
>
> Dear Reviewer 4jUZ,
>
> Thank you for your earlier comments! They were insightful and have helped us strengthen the paper. I just wanted to check if our rebuttal has fully addressed your points. If there are any remaining concerns or areas that would benefit from further clarification or additional supporting results, we would be happy to provide them. We greatly appreciate your feedback and hope that our clarifications have resolved your concerns.
>
> Sincerely,
>
> Authors

---

> ### Comment · Area_Chair_NhjC · 2025-08-07
>
> Dear Reviewer 4jUZ
>
> Please engage in the discussion with the authors and other reviewers as soon as possible.
>
> Thank you.
>
> Best,
>
> AC

---

### Comment · Area_Chair_NhjC · 2025-08-05

Dear Reviewers,

The discussion period deadline is approaching. Please kindly participate to ensure a fair and smooth review process.

Thank you.

Best,

AC

---

### Note · Authors · 2025-08-12

Dear AC and Reviewers,

We thank all reviewers and the AC for their time, constructive feedback, and engagement. We are pleased that three reviewers (VDvq, qNjF, vSDB) expressed support for our work, acknowledging our novelty and convincing results. We are committed to incorporating all suggested revisions into the final version, including: (1) integrating the robustness analyses, (2) updating the computational cost and baseline comparisons, and (3) tempering the statement regarding the geometry prior as recommended.

We also thank Reviewer 4jUZ for their feedback. While unfortunately the reviewer didn’t join the discussion phase, we carefully addressed all points from their review in our rebuttal, including clarifying points regarding dataset usage and literature review, and providing additional experiments to support our claims.

We respectfully request that the AC and reviewers consider the overall support and substantial clarifications provided during the rebuttal and discussion when making the final decision.

Sincerely,

Authors

---

### Decision · Program_Chairs · 2025-09-17

**Decision:**

Accept (poster)

**Comment:**

This paper introduces GeoDiff, a geometry-aware diffusion model designed for reference-driven image completion. The key contribution lies in explicitly incorporating geometric priors—via structure-aware representations and consistency constraints—into the diffusion process, thereby improving the alignment of generated content with the provided reference image. The problem is important for image synthesis and completion, where preserving geometry is crucial, and the proposed method is technically sound and well-motivated.

Reviewers highlighted several strengths. The approach is novel and impactful, addressing a long-standing limitation in diffusion-based image completion by embedding geometric awareness. Empirical results demonstrate consistent improvements over existing baselines, with both qualitative and quantitative evidence supporting the claims. The paper is also clearly written and well-organized, allowing readers to follow the pipeline and contributions effectively.

While some concerns were raised, they were generally minor and have been well addressed during the rebuttal. Reviewer 4jUZ did not participate in any rebuttal discussions, despite multiple reminders from both the AC and the authors. The initial score of 3 is primarily attributed to relatively minor issues in the literature review and benchmark. Such reasoning does not provide a sufficiently convincing basis for a fair overall recommendation. The other reviewers agreed that these points were not critical. Therefore, ACs believe the reviews from Reviewer 4jUZ can be disregarded, and such irresponsible behavior should be flagged.

Overall, this is a timely contribution that advances the state-of-the-art in geometry-aware generative modeling. The novelty, solid empirical performance, and potential impact in downstream tasks such as editing and content creation justify acceptance.